# Rad21 Haploinsufficiency Prevents ALT-Associated Phenotypes in Zebrafish Brain Tumors

**DOI:** 10.3390/genes11121442

**Published:** 2020-11-30

**Authors:** Aurora Irene Idilli, Cecilia Pazzi, Francesca dal Pozzolo, Michela Roccuzzo, Maria Caterina Mione

**Affiliations:** 1Experimental Cancer Biology, Department of Cellular, Computational and Integrative Biology-CIBIO, University of Trento, 38123 Trento, Italy; aurorairene.idilli@unitn.it (A.I.I.); cecilia.pazzi@alumni.unitn.it (C.P.); francesca.dalpozzolo08@gmail.com (F.d.P.); 2Advanced Imaging Facility, Department of Cellular, Computational and Integrative Biology-CIBIO, University of Trento, 38123 Trento, Italy; michela.roccuzzo@unitn.it

**Keywords:** ALT, cohesin, telomeres, zebrafish, brain tumors, DNA damage

## Abstract

Cohesin is a protein complex consisting of four core subunits responsible for sister chromatid cohesion in mitosis and meiosis, and for 3D genome organization and gene expression through the establishment of long distance interactions regulating transcriptional activity in the interphase. Both roles are important for telomere integrity, but the role of cohesin in telomere maintenance mechanisms in highly replicating cancer cells in vivo is poorly studied. Here we used a zebrafish model of brain tumor, which uses alternative lengthening of telomeres (ALT) as primary telomere maintenance mechanism to test whether haploinsufficiency for Rad21, a member of the cohesin ring, affects ALT development. We found that a reduction in Rad21 levels prevents ALT-associated phenotypes in zebrafish brain tumors and triggers an increase in *tert* expression. Despite the rescue of ALT phenotypes, tumor cells in rad21+/− fish exhibit an increase in DNA damage foci, probably due to a reduction in double-strand breaks repair efficiency.

## 1. Introduction

Cohesin is a multi-subunit protein complex, which regulates sister chromatid cohesion during cell cycle [1]. The cohesin complex forms a series of rings around sister chromatids to maintain their pairing during the transition from S-phase to anaphase and to ensure proper segregation of chromosomes. Cohesin is composed of four main subunits (SMC11, SMC3, RAD21, and STAG1 or STAG2), which interact with each other and with accessory proteins that control its loading and unloading onto chromatin [1]. RAD21, the vertebrate ortholog of Saccharomyces Pombe sister chromatid cohesion protein 1 (Scc1), is the core of the complex and links the C-terminus of SMC1 with the N-terminus of SMC3 in the formation of the hinge-ring-like structure of cohesin [2]. RAD21 plays a key role during anaphase because it is the subunit of the cohesin complex that is cleaved by separase [3], thus enabling the separation of sister chromatids. The cohesin complex is widely present on DNA also in the interphase and in postmitotic cells, where, together with the Polycomb group (PcG) silencing proteins [4], it regulates expression of a number of genes involved in development, proliferation, and pluripotency [5]. Moreover, cohesin interacts with the CCCTC-binding factor (CTCF) to organize chromatin in loops and topologically associated domains (TADs) [6,7], self-associating chromosome segments influencing transcriptional activity and 3D genome organization [8].

Another fundamental role of cohesin is related to its involvement in DNA Damage Repair (DDR), where RAD21 is required for the DNA damage-induced G2/M checkpoint and sister chromatid association in homologous recombination-based repair processes [9].

Homologous Recombination (HR) is an essential mechanism involved in genome duplication, telomere maintenance, and repair of DNA double-strand breaks (DSBs) [10]. During repair of DSB, the cohesin complex accumulates near the break site promoting efficient repair by HR and enabling replication by keeping sister chromatids in close proximity to allow template switch. Interestingly, Rad21 mutant cells have been shown to be defective in HR [11], and haploinsufficiency of Rad21 in mice induces impairment of DDR. This demonstrated that Rad21 is critical in promoting HR-mediated DNA damage repair [12].

Alternative Lengthening of Telomere (ALT) is a telomere maintenance mechanism (TMM) that uses HR to maintain telomeres [13]. The activation of a TMM ensures the unlimited proliferative capacity of cancer cells by counteracting telomere shortening. The vast majority of human tumors maintains telomere homeostasis by overexpressing telomerase, a ribonucleoprotein complex, responsible for the synthesis of telomeric DNA using the RNA component (telomerase RNA, *TR* or *TERC*) as a template, reversely transcribed by the catalytic subunit telomerase reverse transcriptase (TERT). Nevertheless, a low percentage (10–15%) of cancers activates the ALT pathway. ALT is prevalent in specific tumor types, such as sarcomas, pancreatic neuroendocrine tumors, and pediatric glioblastomas, which often have poor prognosis [14]. Although the mechanisms beside ALT activation are not well understood, ALT can be identified on the basis of specific markers. Briefly, ALT is characterized by heterogeneous telomere lengths, high levels of telomeric sister chromatin exchanges (T-SCEs) resulting from telomeric crossover events, extrachromosomal C-strand-rich C-circle formation, telomeric DNA nuclear structures called ALT-associated promyelocytic leukemia (PML) nuclear bodies (APBs), and high levels of TERRA expression [15,16]. Moreover, it is suggested that induction of telomeric DSBs can initiate telomere recombination in ALT cells; therefore, ALT telomeres show high levels of DNA damage [16].

In this study, we used a model of progressive juvenile brain tumor [17] that develops ALT as primary telomere maintenance mechanism [18] to test whether haploinsufficiency for Rad21 may modify ALT development. We found that a reduction in Rad21 levels promotes an increase in DNA damage foci, represses ALT phenotype development, and triggers *tert* expression.

Our data suggest that reduced levels of the cohesin member rad21a lead to defects in DSB repair, probably as a consequence of loss of sister chromatid cohesion which also prevents the development of ALT phenotype. Finally, the impairment of the main telomere maintenance mechanism may force these tumors to choose reactivation of telomerase expression as selective pressure for survival.

## 2. Materials and Methods

### 2.1. Maintenance of Zebrafish

Adult zebrafish (Danio rerio) were housed at the Department of Cellular, Computational and Integrative biology (CIBIO) University of Trento and were maintained in the Model Organism Facility under standard conditions [19]. Zebrafish studies were performed according to European and Italian law, D.Lgs. 26/2014, authorization 148/2018-PR to M. C. Mione. Fishes with somatic plasmid expression were generated as described [17,20].

In this study, we used the following zebrafish transgenic lines:*Et(zic4:Gal4TA4, UAS:mCherry)^hzm5^* here called zic:Gal4 [17];*Tg(UAS:egfp-HRASV12G)^io006^* here called UAS:RAS [20];*Rad21a^hi2529Tg/+^* here called Rad21+/− [21].

### 2.2. Genotyping

For *Rad21a^hi2529Tg^* genotyping, we used PCR amplification of genomic DNA with the following primers: rad21 Forward (5′-GCTTGCCAAACCTACAGGTG-3′) and rad21 Reverse (5′-TTTTACGCCCTTAATGAAGTGC-3′), as described in https://zfin.org/ZDB-ALT-041006-8.

### 2.3. Cell Culture and Cell Lines

The U2OS and HeLa cell lines were cultured in DMEM—Dulbecco’s modified Eagle’s medium—supplemented with 10% (*v*/*v*) fetal bovine serum (FBS) in a humidified incubator at 37 °C with 5% CO2. Cell lines were tested regularly for mycoplasma contamination by Celltech CIBIO facility.

### 2.4. Rad21 Immunoblotting

Brains were lysed in RIPA buffer containing protease inhibitors (Complete™, Roche, Basel, Switzerland). Protein concentration was measured using Bicinchoninic Acid (BCA) assay (Pierce, Thermoscientific), and equal amounts of protein were separated by electrophoresis on 10% polyacrylamide gels. Proteins were transferred to Nitrocellulose membranes (Thermoscientific) and blocked using 5% dry milk in TBS containing 0.1% Tween20. The membranes were incubated with rabbit anti-RAD21 antibody [22], diluted 1:5000, and mouse anti-acetylated tubulin (Sigma cat no.T7451), diluted 1:1000. After washing with TBS-Tween, the membranes were incubated with anti-rabbit-HRP (Abcam, cat. No. ab6721) and anti-mouse-HRP (Abcam, cat. No. ab6728) for one hour at room temperature, and visualized with SuperSignal West Dura Extended Duration Substrate (Thermo Scientific, Milan, Italy). Quantification of bands were performed using Image Lab™ Software (Biorad) after background subtraction and followed by normalization on tubulin levels. Data were plotted with GraphPad Prism.

### 2.5. C-Circle Assay and Telomeric qPCR

C-circles assay and telomeric qPCR were performed as described [23] with some modification [18]. Briefly, 30 ng of genomic DNA extracted with a Wizard^®^ Genomic DNA Purification kit (Promega) were used to perform Rolling Circle Amplification (RCA) reaction with and without the addition of Φ29 polymerase (“+/–Φ29”). For dot blot detections, the CCA products (plus 40 μL 2× SSC) were dot-blotted onto 2× SSC-soaked positive nylon membrane, thanks to a 96-well Bio-Dot Microfiltration Apparatus (Biorad). After UV-crosslink and hybridization with a 3′-DIG labeling (CCCTAA)_8_ probe, the membranes were developed as described [23]. Image Lab™ Software (Biorad) was used to analyze dot intensity. The result of the C-circle assay dot blot was evaluated according to [22]. ALT activity was considered significant if at least twice the amount than the levels without Φ29 polymerase. Telomeric qPCR was performed as described [24]. –Φ29 CCircle assay products were diluted four times in water and used as templates in a qPCR using telomF (300 nM) 5′-GGTTTTTGAGGGTGAGGGTGAGGGTGAGGGTGAGGGT-3′ and telomR (400 nM) 5′-TCCCGACTATCCCTATCCCTATCCCTATCCCTATCCCTA-3′ primers. qPCRs using rps11 primers (150 nM) were performed for Single Copy Gene (SCG) normalization. All qPCRs were done in triplicates. Each telomere Ct was normalized with the SCG Ct (normTEL).

### 2.6. Q-TRAP Assay

Real-time quantitative TRAP (Q-TRAP) assay was performed as described [18]. Briefly, protein extracts were obtained adding 200 µL of 1× CHAPS to dissociated brain tumors (RAS and rad21; RAS) and incubate on ice for 30 min. After sample lysate centrifugation (16,000 g for 20 min at 4 °C), total protein concentration was measured using a BCA protein assay kit (Pierce) according to the manufacturer’s protocols. A total of 1 µg of protein extract was used to perform Q-TRAP. A master mix was prepared with 1X SYBR Green Master Mix (Resnova—PCR Biosystem), 100 ng TS primer per sample (5′-AATCCGTCGAGCAGAGTT-3′), 100 ng ACX primer per sample (5′-GCGCGGCTTACCCTTACCCTTACCCTAACC-3′), 1 mM EGTA, and RNase/DNase-free water to a final volume of 25 µL. A total of 2 µL of sample was added to 23 µL master mix in a 96-well PCR plate and incubate 30 min at 30 °C in the dark for extension of telomerase substrate. Real-time PCR was performed with a CFX96 Real-Time PCR Detection System (Bio-Rad) machine using the standard protocol: 95 °C for 10 min; 40 cycles at 95 °C for 15 s, and at 60 °C for 60 s. In all cases, each PCR was performed with triplicate samples and repeated with at least two independent samples. As a negative control, the 1 µg of protein assayed of each sample extract was incubated with RNAse A (QIAGEN) at 37 °C for 20 min. A 1∶10 dilution series of telomerase-positive sample (HeLa) was used for making the standard curve. After PCR, real-time data were collected and converted into Relative Telomerase Activity (RTA) units based on the following formula: RTA of sample = Delta10 ^(Ct sample-Yint)^ /slope. Q-PCR analysis was performed with Microsoft Excel and Graphpad Prism.

### 2.7. TERRA Dot Blot

TERRA dot blot was performed as described in [18]. Briefly, total RNA was extracted with a TRIzol reagent (Invitrogen) and cleaned up using the RNeasy Mini Kit (Qiagen) following the manufacturer’s instructions and treated twice with DNase I (1 unit/μg RNA, Qiagen). A total of 500 ng was dissolved in a total volume of 100 μL of 1 mM EDTA, 50% formamide, and then denatured in a thermocycler at 65 °C for 10 min. As control for DNA contamination, we treated 500 ng of total RNA from each sample with RNAse A (0.2 mg/mL RNAse A, Sigma-Aldrich, for 30 min at 37 °C) prior denaturation. Denaturated RNA was dot-blotted onto 2× SSC-soaked positive nylon membrane and then UV-crosslinker for 3 min/each side. Hybridization was performed at 50 °C O/N with the probe 1.6 kb fragment containing the sequence (TTAGGG)n [25] labeled with the Nick Translation Kit (Sigma-Aldrich) and developed as described above according to [24]. Quantification of dot intensity was performed using Image Lab™ Software (Biorad); after background subtraction and on control normalization. Data were plotted using GraphPad Prism.

### 2.8. Analysis of Gene Expression by qPCR

Total RNA was extracted from juvenile zebrafish brains with the TRIzol reagent (Invitrogen) and cleaned up using the RNeasy Mini Kit (Qiagen) following the manufacturer’s instructions and with two treatments with DNase I (1 unit/μg RNA, Qiagen). The RNA concentration was quantified using nanodrop2000 (Thermo Fisher), and the VILO superscript KIT (Thermo Fisher) was used for First-strand cDNA synthesis according to the manufacturer’s protocol. qRT-PCR analysis was performed using the qPCR BIO Sygreen Mix (Resnova—PCR Biosystem) using a standard amplification protocol. The primers used are the following: zebrafish tert forward 5′-CGGTATGACGGCCTATCACT-3′ and reverse 5′-TAAACGGCCTCCACAGAGTT-3′; zebrafish Rad21a forward: 5′-CTTTCGCTCTTGAGCCCATC-3′ and reversed 5′-GAGCAGGCAATGAGAAGAGC-3′; zebrafish rps11 forward: 5′-ACAGAAATGCCCCTTCACTG-3′ and reverse:

5′- GCCTCTTCTCAAAACGGTTG-3′; data analysis was performed with Microsoft Excel and Graphpad Prism. In all cases, each qPCR was performed with triplicate samples and repeated with at least two independent samples. Data are expressed as fold changes compared to controls.

### 2.9. Immunofluorescence Combined with Quantitative Fluorescence In-Situ Hybridization (Q-FISH) on Interphase Nuclei

The protocol used for immunofluorescence combined with the QFISH was the same described in [18]. Briefly, cell suspensions derived from zebrafish brain tumors were seeded on Poly-lysine (1 µg/mL) (Sigma-Aldrich) slides. After 1 wash in TBS 1× per 5 min, slides were fixed for 10 min in 2% paraformaldehyde, 2% sucrose at RT, and then washed twice in TBS, followed by permeabilization for 15 min with 0.5% Triton. After 3 washes in TBS 1×, slides were incubated 1 h at RT in blocking buffer (0.5% BSA, 0.2% Gelatin cold water fish skin in 1× PBS) and then incubated overnight at 4 °C with γH2AX primary antibody 1:300 (Millipore, cat. No05-636). The following day, after three washes in blocking buffer, slides were incubated with secondary antibody 1:500 for 2 h at RT (goat-anti-mouse 488—Thermo Fisher). Then, the slides were washed 3 times in 1× TBS (5 min each) and 1 time in Q-FISH washing buffer (0.1% BSA, 70% formamide, 10 mM Tris pH 7.2). Telomere FISH Hybridization was performed with the PNA TelC-Cy3 probe (PANAGENE) as previously described [26]. Nuclei were counterstained with DAPI. Z-stacks Images were captured at 100× magnification (Plan Apochromatic 100×/1.45 oil immersion objective) using an inverted Nikon Ti2 fluorescent microscope equipped with a monochromatic Andor Zyla PLUS 4.2 Megapixel sCMOS camera. Images were processed for background subtraction using a custom macro designed with the Fiji/ImageJ software. Colocalization analysis was performed with DiAna (ImageJ) [27], calculating co-localization between objects in 3D, after 3D spot segmentation.

### 2.10. Statistics

All the graphs and the statistical analysis were generated and calculated using the GraphPad Prism software version 5.0. All the graphs represent mean +/− SD and were analyzed with Mann–Whitney—nonparametric statistical test (no Gaussian distribution, two-tailed, an interval of confidence: 95%) except for different details reported in the figure legends.

## 3. Results

### 3.1. Rad21a Haploinsufficiency Does Not Worsen Brain Tumor Development

Here we use a zebrafish juvenile brain tumor model that progressively develops ALT [18] to test whether haploinsufficiency for Rad21, a member of the cohesin complex, may impair ALT development. To generate a brain tumor model with reduced levels of rad21, we used the *rad21a^hi2529Tg/+^* line, created by retroviral-mediated insertional mutagenesis [21] with the insertion mapped in the first intron of the *rad21a* gene, in chromosome 16. The rad21;RAS brain tumor model was generated by first crossing the zic driver line *(Et(zic4:GAL4TA4,UAS:mCherry)^hmz5^)* with *rad21a^hi2529Tg/+^* fish, and named zic:rad21a+/−, followed by the injection of the human UAS:eGFP-HRAS^v12^ oncogene into one-cell-stage embryos (Figure 1A), or by crossing zic:rad21a+/− fish with the *tg(UAS:eGFP-HRASV12G)^io006^* line to generate fish with brain tumors and *rad21a* haploinsufficiency.

As bi-allelic mutations of *Rad21* in mouse [28] and *rad21a* in zebrafish (https://zfin.org/ZDB-GENE-030131-994) lead to embryonic lethality, we used *zic:rad21a^hi2529Tg/+^* heterozygous to generate tumor models with reduced rad21a levels (named rad21;RAS).

Brain tumor masses develop with the same frequency in RAS and rad21;RAS fish and appear as diffusely infiltrating malignant masses located in the telencephalon, diencephalon, and cerebellum (Figure 1A). Both RAS and rad21;RAS fish developed tumors with similar timing and overall survival (Figure 1B).

To confirm that *rad21a^hi2529Tg/+^* fish were haploinsufficient for rad21a, we quantified the expression of *rad21a* through RT-qPCR in brains and tumors from rad21+/+ and rad21+/− brains. *Rad21a* transcript levels were significantly decreased in rad21+/- compared to control brains, and in rad21+/-;RAS compared to RAS tumors, suggesting that the insertion present in one allele of *rad21a^hi2529Tg/+^* fish leads to reduced expression of *rad21a* (Figure 1C). Moreover, analysis of Rad21 protein levels in the zebrafish brain showed a reduced level of Rad21 in both control and tumors from *rad21a^hi2529Tg/+^* fish (Figure 1D, Appendix A).

### 3.2. Rad21a Haploinsufficiency Prevents ALT

ALT relies on HR to maintain telomere length [29]; cohesin complex and its subunits can promote efficient HR by promoting sister telomere cohesion. To study whether a reduction in *rad21a* expression affects ALT development, we performed an analysis of TMM in RAS and rad21;RAS brain tumors.

We first evaluated telomere hetereogenety, one of the most common ALT features. Telomere qPCR analyses using genomic DNA extracted from RAS and rad21;RAS tumor samples and from control brains revealed an increase of telomere content heterogeneity in the RAS samples, but similar-to-control telomere content in rad21;RAS tumors (Figure 2A). Then, we sought to evaluate differences in telomere length distribution at single-cell resolution, by performing Q-FISH experiments using a fluorescently labeled telomere specific probe.

Q-FISH in rad21;RAS tumor cells did not show the ultrabright foci seen in RAS tumors (Figure 2B, white arrows). Moreover, the increase in number of individual telomeric foci per nucleus (Figure 2C) suggested a loss of telomere clusters, which are hallmarks of ALT [30], in favor of single telomere signals. In addition, rad21;RAS tumor cells exhibited a decrease of mean signal intensity (Figure 2D), telomere length heterogeneity (Figure 2E), and ultra-long telomeres (Figure 2F) compared to RAS tumor cells. These quantitative analyses carried out on Q-FISH signals suggested that rad21;RAS tumors may not show the ALT phenotype.

Next, we measured the abundance of ALT telomeric recombination products, C-circles. We performed C-circles assay followed by telomeric dot blot (Figure 2G) in zebrafish control brains, tumor samples, and, as additional controls, in ALT+ U2OS and telomerase+ HeLa cell lines [23,30]. Quantification of the signals obtained across three biological replicates indicated that C-circles accumulate in RAS cells, but not in rad21;RAS tumor cells (Figure 2H).

We investigated the expression of the telomeric lncRNA TERRA, an RNA polymerase II transcript produced from subtelomeric regions towards chromosome ends [31]. TERRA was found to be expressed at high levels in ALT cancer cells [32,33] and we previously found that zebrafish ALT brain tumors show increased TERRA expression [18]. RNA dot blot analysis on total RNA extracted from control, RAS or rad21;RAS brain samples showed that the levels of TERRA in RAS Rad21+/− tumors were similar to those found in control brain and in telomerase+ HeLa cells, much lower than those present in ALT brain tumors (RAS) and U2OS cells (Figure 2I,J, Appendix A).

Thus, rad21a haploinsufficiency is preventing the development of several ALT markers and the increase of TERRA expression in zebrafish brain tumors.

### 3.3. Rad21;RAS Brain Tumors Show an Increase in γH_2_AX TIFs, but Have Normal Tert Levels and Rad51 Foci

In telomerase-positive cells high levels of oncogene-induced telomeric DNA damage could initiate telomere recombination and trigger ALT features [34]. RAD21 is believed to function in sister chromatid alignment as part of the cohesin complex. By holding sister chromatids together, RAD21 and the cohesin complex can facilitate the repair of DNA damage incurred during DNA replication [11,12]. Therefore, we asked whether rad21a haploinsufficiency, which prevents ALT, was accompanied by a change of telomeric DNA damage, possibly through DNA damage repair by HR. We assessed DNA damage by measuring the level of the DNA damage marker γH2AX, and the occurrence of HR using immunofluorescence for the DSB repair protein RAD51, both combined with telomeric FISH, in RAS and rad21;RAS brain tumors. Remarkably, we found that while an accumulation of γH2AX foci was present at telomeres (telomere induced foci, TIFs) in RAS tumors [18], rad21;RAS brain tumor cells showed an even greater number of γH2AX-positive TIFs (Figure 3A,B). We did not find a significant difference in telomeric Rad51 foci between RAS and rad21;RAS tumors (Figure 3C,D), thus suggesting that the reduced levels of rad21a do not change the proportion of double-strand breaks repaired by Rad51 loading.

Our recent study on ALT zebrafish brain tumors showed that the development of ALT occurs during brain tumor development upon downregulation of *tert*, and activation of ALT can be prevented by the overexpression of telomerase [18]. Therefore, we checked the expression of *tert*, the catalytic subunit of telomerase, in rad21;RAS brain tumors using qPCR. We found that, unlike RAS tumors where *tert* is downregulated, rad21;RAS tumors expressed levels of *tert* similar to control brains (Figure 3E). When we measured telomerase activity using TRAP assay, we found that rad21;RAS brain tumors showed a trend versus telomerase reactivation (Figure 3F). These results suggest that the reduction of *rad21a* levels prevents the downregulation of *tert* expression seen in zebrafish RAS tumors and may promote telomere maintenance by telomerase activity.

Taken together, these observations suggest that the reduction of rad21a levels leads to a decrease of DSB repair efficiency, and, at the same time, relieves the transcriptional repression on tert in previous ALT+ tumors; thus, allowing telomere maintenance of tumor proliferating cells through other mechanisms, either through a partly restored telomerase dependent mechanisms, or through telomerase- and ALT-independent mechanisms [35], or both.

## 4. Discussion

Reduction of cohesin levels have profound consequences on telomere homeostasis. Here, we studied the effects of a reduction of rad21a levels, a cohesin subunit, on telomere maintenance in ALT brain tumors. The activation of a telomere maintenance mechanism is an important step towards immortalization of cancer cells, and a number of juvenile brain tumors adopt ALT, rather than telomerase re-expression, to promote telomere manteinance and lengthening. ALT relies on HR, thus it may be affected by a reduction of cohesin functions, which promote HR by maintaining sister chromatid cohesion. Our data indicate that the reduction of rad21a levels has marked effects on TMM in brain tumor cells in vivo, as it prevents ALT-associated phenotypes and restores normal levels of TERRA and *tert* expression.

Another important finding of our study is the further increase of DSBs in Rad21 haploinsufficient brain tumors compared to ALT brain tumors [18]. Besides the canonical role in mitosis, the ability of the cohesin complex to establish sister chromatid cohesion facilitates the repair of DNA damage and confers the spatial proximity required for the rejoining of DSBs. Moreover, strand invasion and template exchange with sister chromatids in DNA homology repair (DHR) is favored by holding sister chromatids together near DBS sites [36,37]. The Rad21 subunit is believed to play a dual role in DHR, both in the modulation of the alignment of sister chromatids and by favoring homologous recombination-mediated DBS repair [38,39]. We propose that dysregulation of the Rad21 component of the cohesin complex might promote genomic instability with loss of sister chromatid cohesion that impedes the homologous recombinational repair of DNA damage. Since the proximity and separation between two homologous telomere ends are essential for the ALT telomeric homologous recombination mechanism, it prevents the ALT phenotype and induces tert re-expression to restore a telomere maintenance mechanism as selective pressure for tumor survival.

During repair, a local activation of separase, necessary to cleave rad21, has been shown to promote DSB repair in mammalian cells [40]. Moreover cohesin removal by WAPL (a cohesin accessory protein) was found to be necessary for RAD51-mediated restart of stalled replication forks [41]. Thus, cohesins allow repair of DSB and restart of stalled replication forks on one side by loosening local cohesion through WAPL and separase, necessary to provide space for Rad51-mediated repair through HR, on the other side by maintening cohesion ahead and behind of DSBs.

What is the relation between DSBs and ALT? In specific subpopulations of cancer cells, ALT is initiated by the induction of DSBs and by replication stress, which promote telomere recombination. ALT cells seem to adopt a specialized homologous recombination dependent pathway that involves both intra- and inter-telomeric recombination and replication [42]. Indeed, ALT has been associated with two distinct break-induced replication (BIR) pathways [43], one dependent on Rad52 (which promotes the invasion of single-stranded telomere ends into telomeric dsDNA in the presence of RPA), while the other may rely on Rad51 (which has the ability to promote the invasion of single-stranded telomere ends when RPA is absent). Our data on the increase of DSBs without a corresponding increase in rad51 loading in Rad21+/−;RAS brain tumors in which ALT is prevented, suggest that the reduction of rad21 levels prevent signalling to wapl and separase for rad51 loading. ALT in these tumors may be linked to rad52-dependent BIR. In this case, the increase of DSB found in rad21 haploinsufficient brain tumors may not promote ALT because the reduction of rad21a levels impairs cohesion and does not provide the spatial proximity required for recombination.

A third hypothesis on the effects of rad21a haploinsufficiency in preventing ALT in brain tumors is based on its role in chromatin organization and transcription. Dysregulation of the cohesin complex might promote genomic instability by perturbing chromatin interactions [44]. The cohesin complex, in association with CTCF, is required for transcriptional insulation in vertebrate cells [5], and cohesins can associate with a number of active promoters also in a CTCF-independent way [45] to regulate gene transcription and promoter–enhancer interactions [46]. Ing-Simmons et al. [47] found that most human chromosomes have a major CTCF/Rad21 binding site within 1–2 kb from the telomeric repeats, corresponding to TERRA promoter sequences. Indeed, depletion of CTCF/Rad21 led to a marked reduction of TERRA transcription at telomeres containing promoter proximal CTCF/Rad21 binding sites. Moreover, depletion of CTCF and Rad21 caused a three-fold increase in γH2AX and 53BP1-associated TIFs by telomere uncapping, due to the loss of shelterin deposition that occurs when CTCF/cohesin barrier functions are impaired and subtelomeric nucleosomes may obstruct shelterin assembly at the telomere repeat tracts [48].

Recently, it was found that CTCF/cohesin complexes preferentially bind hypomethylated CpG island [49]. In contrast to most genes, TERT promoter hypomethylation could be associated to its transcriptional repression in cancer [18,50], suggesting that tert promoter sequences may be a binding site for the CTCF/cohesin complex. A reduction of rad21a levels may therefore loosen the organization of the regulatory elements in this region and allows transcription of *tert*.

## 5. Conclusions

In conclusion, our data suggest that a reduction in the levels of rad21a has profound consequences on the homeostasis of the telomeric region of zebrafish chromosomes and can tip the balance of the TMMs adopted by cancer cells towards very different scenarios. This adds further complexity in addressing the importance of co-occurring mutations and haploinsufficiency in brain tumors as they can modulate telomere maintenance mechanisms, and, therefore, cancer progression, cell heterogeneity, and drug sensitivity.

## Figures and Tables

**Figure 1 genes-11-01442-f001:**
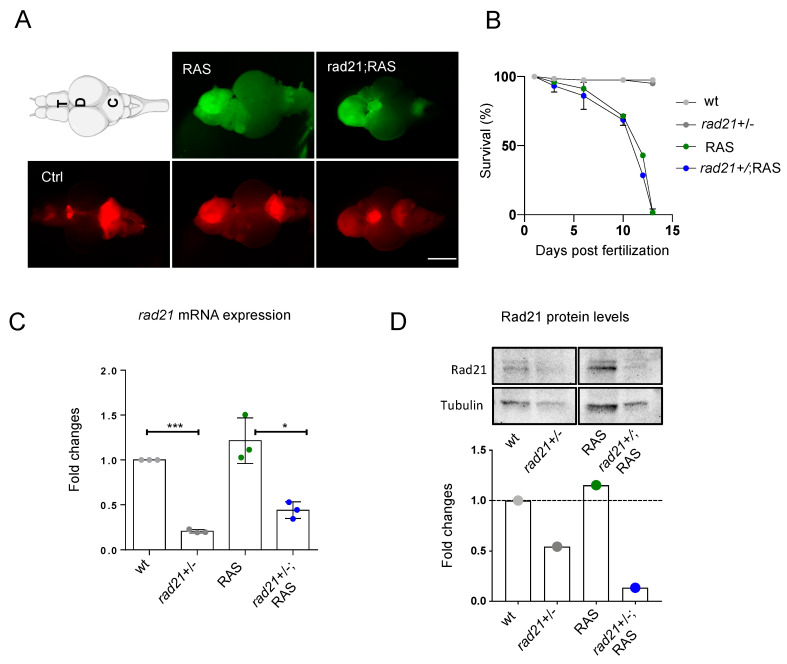
Rad21 heterozygous fish develop brain tumors with the same frequency and location than wt. (**A**) Representative images of zebrafish RAS and rad21;RAS brains used for this study. The expression of UAS:eGFP-HRAS^V12^ was induced in a population of brain progenitor cells using the driver line zic:GAL4 (visualized through mCherry expression, bottom panel). Tumor formation is visualized through eGFP expression in 1-month-old fish (upper panel). The location of tumors appears to be similar in both RAS and rad21;RAS fish. Scale bar: 0.5 mm. T: telencephalon; OT: optic tectum; C: cerebellum. (**B**) Survival curve of RAS and rad21;RAS shows a very low survival rate (approx. 1%) for both genotypes n = 2 in control groups (gray dots) and n = 4 in tumor groups (blue and green dots). (**C**) Expression of zebrafish *rad21a* mRNA in controls and brain tumors measured by RT-qPCR. Values were normalized to *rps11* mRNA levels; n = 3 in all groups; *** *p* < 0.001; * *p* < 0.05. (**D**) Rad21 protein levels in Western blot and relative quantification. Rad21 levels are reduced in control (2nd lane) and tumor brains (4th lane) from *rad21a^hi2529Tg/+^* fish compared to the wild type (fold changes). Values were normalized to *tubulin* levels.

**Figure 2 genes-11-01442-f002:**
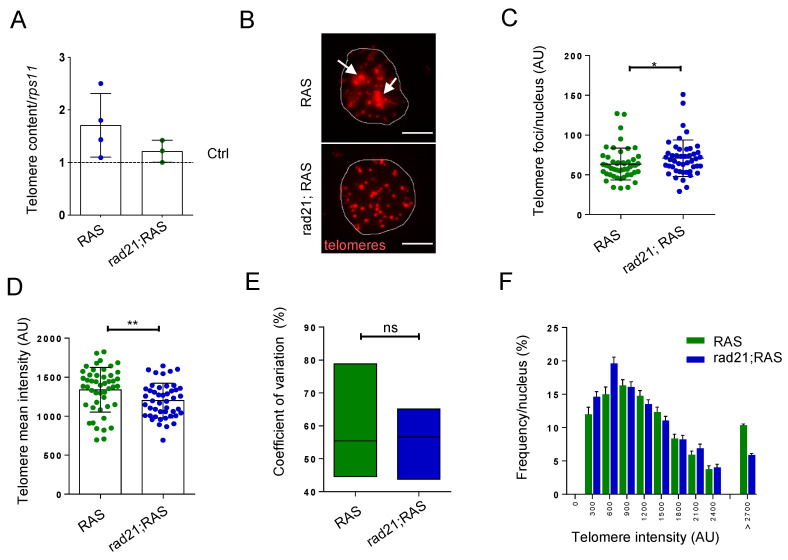
Rad21 depletion prevents ALT, normalize telomere length and reduces TERRA levels in brain tumors. (**A**) Telomere length analysis through telomeric qPCR in RAS, RAS Rad21 +/− brain tumors compared to control brains (dashed lines). The relative amount was normalized to the signal of a single copy gene (*rps11*). Each dot represents one sample, n = 3 in all groups. (**B**) Representative fluorescent microscope images of Q-FISH analysis of RAS and rad21;RAS nuclei with ultrabright foci (white arrows). Scale bar: 5 µm. (**C**) Quantification of the numbers of telomere foci per nucleus and (**D**) quantification of relative telomere length intensity measured by Q-FISH in RAS and rad21;RAS brain tumors. * *p* < 0.05, ** *p* < 0.01. RAS n = 48 nucleus; rad21;Ras n = 46 nucleus. (**E**) Analysis of coefficient of variation of telomere mean intensity measured in (**C**) between RAS and rad21;RAS. (**F**) Distribution of telomere length evaluated by Q-FISH (**C**) in RAS and rad21;RAS tumors. The signals >2700 AU could represent ultra-long telomeres or telomeric clusters, an ALT feature. (**G**) Quantification analysis of C-circle assay dot blots (each dot represents one sample). Determination of C-circles amount was calculated after subtracting global background and specific –θ29 signal. AU: arbitrary unit. (**H**) Representative C-circle assay followed by telomeric dot blot in control, RAS and rad21;RAS brain tumors compared with telomerase+ HeLa cells and ALT+ U2OS cells. Reactions without phy29 polymerase (–θ29) were included as a control. (**I**) Quantification of TERRA expression measured by dot blot using total RNA (500 ng) of control, RAS, and rad21;RAS brain tumors. As positive and negative control, U2OS and Hela cells were added. (**J**) Representative RNA dot blot hybridization using total RNA (500 ng) of control, RAS, and rad21;RAS brain tumors. As positive and negative control, U2OS and Hela cells were added. For (**G**,**I**), a minimum of three independent experiments were performed and analyzed with 1 way anova Kruskal–Wallis—non-parametric test.

**Figure 3 genes-11-01442-f003:**
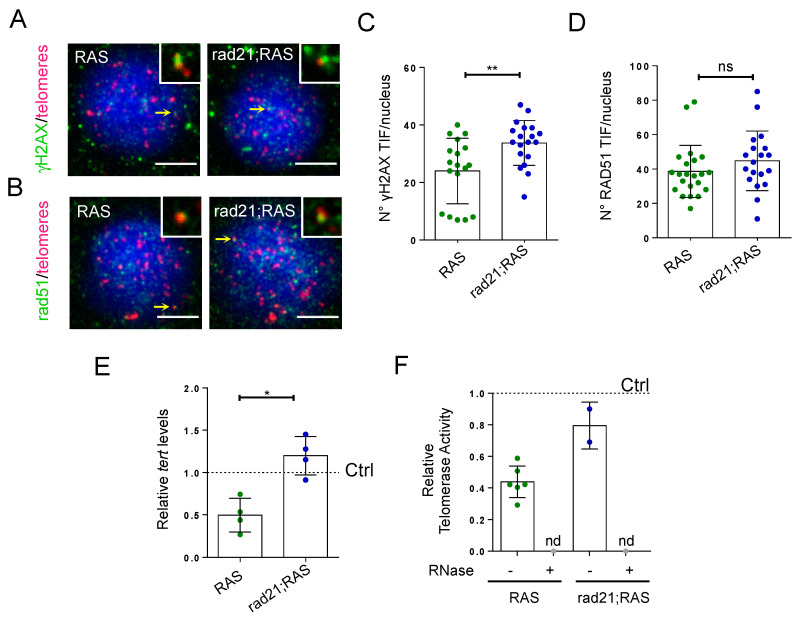
Rad21 depletion increases *tert* expression and γH2AX TIFs in brain tumors. (**A**) Representative fluorescent microscope image of Q-FISH and γH2AX immunostaining in RAS and rad21;RAS tumor cells. Scale bar: 5 µm. Insets show examples of co-localization of the Q-FISH and antibody signals, in foci pointed by the yellow arrows. (**B**) Representative fluorescent microscope image of Q-FISH and Rad51 immunostaining in RAS and rad21;RAS tumor cells. Scale bar: 5 µm. Insets show examples of co-localization of the Q-FISH and antibody signals, in foci pointed by the yellow arrows. (**C**) Quantification of the numbers of telomere foci which are positive for γH2AX immunostaining (nuclei = 27), also known as TIFs, in RAS and rad21;RAS brain tumors cells. Each dot represents a nucleus. ** *p* < 0.01. (**D**) Quantification of the numbers of telomere foci which are positive for rad51 immunostaining (RAS nuclei = 22; rad21;RAS nuclei =19) in in RAS and rad21;RAS brain tumors cells. ns = not significant. Each dot represents a nucleus. (**E**) Expression of zebrafish *tert* mRNA in brain tumors measured by RT-qPCR. Values were normalized to *rps11* mRNA levels and are relative to *tert* expression in control brains (dashed line set at 1.0). * *p* < 0.05. Each dot represents a sample. (**F**) Relative telomerase activity measured by Q-TRAP in control, RAS, and rad21;RAS brains, using 1 μg of protein extracts. RNase treatment (+) was used as a negative control to confirm the specificity of the assay; each dot represents one sample.

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
