# Peer review of "Rad21 Haploinsufficiency Prevents ALT-Associated Phenotypes in Zebrafish Brain Tumors"

_genes, 2020, doi:10.3390/genes11121442_

Round 1

Reviewer 1 Report

The authors have clarified all the point raised and have substantially improved the quality of the manuscript. The paper is now suitable for publication on Genes Journal 

Author Response

Thank you.

Reviewer 2 Report

The authors have addressed the majority of my concerns adequately, but I still have reservations about the results and their interpretation.

  1. It must be stressed that the definition of ALT is telomere extension/telomere maintenance in the absence of telomerase. The fact that the authors still see telomerase activity in RAS tumors via qTRAP makes them telomerase positive, not ALT. It is plausible that these tumors may be heterogeneous (some telomerase positive, some ALT positive), but the authors disagree with this. While I agree that the increase in telomere content (PCR cannot measure length) of RAS-only tumors is interesting, it does not indicate ALT, as telomerase activity is still present. The authors previous publication also demonstrated a heterogenous decrease in telomere length in RAS tumors (you would expect an increase in ALT tumors), and did not check maintenance over time. These results suggest that the decreased telomerase activity is causing a decrease in length, not that ALT was activated. At best, the authors can claim a change in ALT associated phenotypes and telomerase activity, but not ALT itself.
  2. The results in Fig. 1D are unconvincing as the decrease in rad21 matches the decrease in the loading controls. The authors need to include a better WB demonstrating a change, as the rest of their study depends on this result.

Author Response

The authors have addressed the majority of my concerns adequately, but I still have reservations about the results and their interpretation.

  1. It must be stressed that the definition of ALT is telomere extension/telomere maintenance in the absence of telomerase. The fact that the authors still see telomerase activity in RAS tumors via qTRAP makes them telomerase positive, not ALT. It is plausible that these tumors may be heterogeneous (some telomerase positive, some ALT positive), but the authors disagree with this. While I agree that the increase in telomere content (PCR cannot measure length) of RAS-only tumors is interesting, it does not indicate ALT, as telomerase activity is still present. The authors previous publication also demonstrated a heterogenous decrease in telomere length in RAS tumors (you would expect an increase in ALT tumors), and did not check maintenance over time. These results suggest that the decreased telomerase activity is causing a decrease in length, not that ALT was activated. At best, the authors can claim a change in ALT associated phenotypes and telomerase activity, but not ALT itself.

We change the text as the reviewer suggested (see track tracing in the new version of the ms) and we are now stating that Rad21 haploinsufficiency prevents ALT associated phenotypes. To clarify we would like to point out that in our previous publication we demonstrated an increase in telomere content (by telomere qPCR) AND an increase in telomere length and heterogeneity (by QFISH and TRF) in RAS tumors compared with control brain (Idilli et al., 2020, PMID: 32331249) as would be expected in ALT, and not a decrease, as the reviewer says.

  1. The results in Fig. 1D are unconvincing as the decrease in rad21 matches the decrease in the loading controls. The authors need to include a better WB demonstrating a change, as the rest of their study depends on this result.

We agree with the reviewer that the western blot in figure 1D is not perfect, therefore to document the reduction of Rad21 protein levels in rad21+/-  fish, we quantified the rad21 bands and normalize them to the tubulin content. Quantification graph is now shown. The decrease in Rad21 levels in rad21 heterozygous samples is now clear, and adds up to the reduction of rad21 mRNA  shown in fig. 1C.

This manuscript is a resubmission of an earlier submission. The following is a list of the peer review reports and author responses from that submission.

Round 1

Reviewer 1 Report

In the paper entiled "Rad21 haploinsufficiency prevents ALT development
 in zebrafish brain tumors" the authors show the effects of RAD21 knock down on the switch between ALT and telomerase mechanisms of telomere maintenance. While it is enough clear that rad21 loss of function decreases ALT activity, hTERT mRNA level dosage is not sufficient to claim a shift between ALT and hTERT mechanism. Interestingly, in this model system the authors do not see any effect on tumor growth, which could be coherent with a switch between ALT and hTERT, but, to be fully convincing, they have to measure hTERT catalytic activity by TRAP assay (it can be done also by qPCR). I think that the authors have to present also the data about tumor mass growth and overall survival in RAS and RAS; RAD21+/-, which are cited but not shown in figure. Paradoxically, some brain tumors (in humans) have been reported to grow also in absence of any telomere maintenance mechanism, so it is not obvious that ALT suppression induces hTERT activity. Another point that needs to be improved is the analysis of DDR. First of all it is not clear why the authors expected a difference in DDR activation between rad21 +/+ and +/- cells in RAS activated background. However, the DDR increase upon RAD21 knock down is left not interpreted by the authors and has nothing to do with hTERT (which instead is known to protect from DDR). Unless the authors decide to deeply study the mechanisms by which RAD21 could increase DDR, I suggest to remove these data that do not add much to the paper conclusions. To enforce the message, the authors could replace this figure with the analysis of the presence of APBs at telomeres, which are markers for ALT activity. 

Reviewer 2 Report

This manuscript tests the hypothesis that rad21 haploinsufficiency prevents the development of ALT as a TMM in Zebrafish RAS brain tumor model. The manuscript then goes on to examine a number of ALT associated phenotypes (telomere heterogeneity, C-circles, TERRA levels, tert expression, etc.) and go on to show that these phenotypes are reduced in rad51;RAS tumors. The techniques are performed well, although I have some concerns regarding sample size.

The fundamental issue I have with this manuscript is the assumption that ALT associated phenotypes are direct measures of ALT activation/activity. ALT is defined as telomere elongation/maintenance in the absence of telomerase. Each one of these phenotypes can occur in telomerase cells and in TMM negative cells.

Major comments.

1. The strict definition of ALT is telomere extension/telomere maintenance in the absence of telomerase. The authors observed lower levels of ALT associated phenotypes in rad21;RAS tumors, but ALT associated phenotypes do not always correlate to ALT activation/activity. The authors must show the maintenance of telomere length in RAS tumors without telomerase activity over a period of time, and the maintenance of telomere length with telomerase activity over time in rad21;RAS tumors. This can be done with qTRAP and RT-PCR for telomere length (although preferably with a TRF). Without this data the authors can only state that rad21 insufficiency reduces the formation of ALT associated phenotypes (not ALT activation/activity) in zebrafish RAS brain tumors.

2. If the tumors are thought to be heterogeneous (some telomerase positive, some ALT positive) and the results indicate a shift in the population to more ALT positive cells in rad21;RAS tumors, then the authors must demonstrate this by culturing clones and testing telomere maintenance and telomerase activity over time.

Minor comments

1. mRNA levels do not always correlate with protein expression levels. Could the authors please provide a western blot showing rad41a expression to supplement the RT-qPCR in figure 1b?

2. Could the authors please clarify how many sample sizes and what was counted in the figure legends. This is not always clear.

3. Figure 2 has a panel labelled L which is not taken account of in the figure legend or manuscript text.

4. Could the authors please provide the unedited dot blots used in figure 2H and 2L as a supplementary figure?

5. Do the RAS tumors develop APBs like mammalian ALT tumors? If so, this would be another good ALT associate phenotype to examine in this model.

6. Could the authors please provide examples of what they are calling co-localizations in figure 3A and 3C?

7. Could the authors please perform and comment on significance tests between groups in Figure 3G and 3I? In addition, the authors must include at least 3 biological replicates for these figures, 2 independent experiments are not enough.